# Regulation of Plasmodesmata Function Through Lipid-Mediated PDLP7 or PDLP5 Strategies in *Arabidopsis* Leaf Cells

**DOI:** 10.3390/plants15010145

**Published:** 2026-01-04

**Authors:** Xin Chen, Ning-Jing Liu, Jia-Rong Hu, Hao Shi, Jin Gao, Yu-Xian Zhu

**Affiliations:** 1State Key Laboratory of Gene Function and Modulation Research, School of Life Sciences, Peking University, Beijing 100871, China; xin-chen@pku.edu.cn (X.C.); 2501110516@stu.pku.edu.cn (J.-R.H.); shihao@pku.edu.cn (H.S.); 2School of Life Sciences, East China Normal University, Shanghai 200241, China; liuningjing1@yeah.net; 3Dalian Institute of Chemical Physics, Chinese Academy of Sciences, Dalian 116023, China; gaojin0903@dicp.ac.cn; 4Institute for Advanced Studies, Wuhan University, Wuhan 430072, China

**Keywords:** plasmodesmata, PDLP7, membrane order, PDLP5, sphingolipids, plant immunity

## Abstract

Plasmodesmata (PDs) are enriched in sphingolipids and sterols, creating a specialized environment for regulatory proteins like plasmodesmata-localized proteins (PDLPs). How PDLPs regulate PD function in a specific lipid environment remains poorly understood. Here, we provide a unique insight from the interaction network of two different PDLPs together with sphingolipids and propose a concept that PDLPs form homo- or hetero-dimers only in the presence of sphingolipids. Located in the detergent resistance region, PDLP7 demonstrated the ability to influence the sphingolipid composition in PD-enriched fraction, particularly the GIPC content, and finally, modulating the membrane order. The presence of sphingolipids, in turn, affected the oligomeric state of PDLP7 in membranes. The PDLP7 recombinant protein existed as a monomer in vitro, but it formed self-aggregates in yeast and plant cells. We further examined PDLP5, another known phytosphinganine (t18:0)-specific binding PDLP, alongside PDLP7, and confirmed a similar interaction pattern: no direct interaction was observed in vitro, but interactions were noted in vivo. Co-overexpression of the two disrupted their PD localization and induced the upregulation of *pathogenesis-related protein 1* (*PR1*). In summary, we gained insights into the network of PDLPs with lipids and propose that PDLPs were under precise regulation during plant development and stress responses.

## 1. Introduction

Plasmodesmata (PDs) are specialized membrane-lined channels that maintain symplastic continuity between adjacent cells [1]. Structurally, PDs consist of a continuous plasma membrane (PM) traversing the cell wall and a central constricted strand of the endoplasmic reticulum (ER), termed the desmotubule [2,3]. The ER and PM are interconnected by specialized proteins, forming a cytoplasmic sleeve between them that allows the passage of molecules [4]. Functionally, PD mediate the selective trafficking of signaling molecules, metabolites, and macromolecules such as proteins and RNA, acting as pivotal regulators of plant growth, development, and responses to environmental stresses [5,6,7,8]. In recent decades, many studies have shown that PD permeability is dynamically modulated through several key mechanisms, most notably callose turnover at the neck region [9,10], changes in membrane lipid composition [11], and the action of proteins at ER/PM contact sites [12], together enabling precise control of intercellular transport.

Compared with the bulk PM, PD membranes display unique compositional features, being highly enriched in sphingolipids and sterols [11]. Such lipid enrichment leads to the formation of ordered nanodomains, often referred to as lipid rafts [13,14], which provide a specialized biochemical environment within PD [14]. For instance, sterol–sphingolipid interactions facilitate nanodomain formation in the outer PM leaflet, enabling the recruitment of glycosylphosphatidylinositol (GPI)-anchored proteins [11]. Likewise, sterols and phosphatidylinositol 4-phosphate (PI4P) promote nanodomain assembly in the inner leaflet, hosting proteins such as Remorins that modulate lipid order and domain organization [15]. Furthermore, perturbations in sphingolipid biosynthesis can alter PD architecture, including changes in cell wall thickness, underscoring the functional importance of lipid-driven nanodomains in PD regulation [16]. However, while many studies have shown that lipids govern protein localization and function at PD, the mechanisms by which PD-associated proteins modulate PD lipid composition remain poorly understood.

Among PD-associated proteins, the plasmodesmata-located proteins (PDLPs), a family of eight members first identified through a proteomic survey of *Arabidopsis* suspension culture cells, have emerged as key regulators of PD function [17]. Investigations of PDLP1 and PDLP5 have established their involvement in plant immunity, which is accomplished by PD-callose and/or lipid-mediated processes that facilitate various aspects of cellular biology in plants [18,19]. As non-enzymatic plasmodesmal regulators, most PDLPs exert their biological functions through interactions with other proteins or small molecules, such as AZI1 (lipid transfer-like protein required for AzA- and G3P-induced SAR) [20], Acyl-CoA-binding protein 6 (ACBP6) [21], *GLUCAN SYNTHASE-LIKE* (GSL8) [22], and NON-RACE SPECIFIC DISEASE RESISTANCE/HIN1 HAIRPIN-INDUCED-LIKE protein 3 (NHL3) [23]. Our previous work also showed that PDLP5 specifically binds phytosphinganine (t18:0)-based sphingolipids—lipid species enriched at PDs—via a conserved transmembrane domain motif, thereby promoting its localization to PDs, enhancing callose deposition, reducing PD aperture, and ultimately conferring increased resistance to both bacterial and fungal pathogens [24]. In addition, PDLP7 was recently found to be transcriptionally upregulated during viral infection and to interact with glucan endo-1,3-β-glucosidase 10 (BG10), potentially blocking BG10-mediated callose degradation, resulting in callose accumulation at plasmodesmata and enhanced viral resistance [25]. Together, these findings highlight PDLP5 and PDLP7 as key modulators of PD aperture through their regulation of callose metabolism and lipid interactions. However, how two different yet similar proteins evolve distinct functions to support plant growth and development, whether they function redundantly or antagonistically, and how they interact with lipid nanodomains at PDs remain poorly understood.

Given the central role of distinct PDLP family members in modulating membrane organization, we hypothesize that PDLP7 interacts with specific plasmodesmata lipids to fine-tune membrane architecture and works in coordination with another lipid-binding PDLP, PDLP5, to regulate intercellular communication. To test this integrated hypothesis, we employed a suite of molecular and microscopic approaches, defining the mechanisms by which these two closely related PDLPs act in a coordinated yet tissue-dependent manner to modulate membrane organization and balance plant development with immune responses.

## 2. Results

### 2.1. PDLP7 Is Located in the PD Lipid Rafts and Also Regulates PD Sphingolipids

Previously, we demonstrated that PDLP7 can bind to t18:0 through surface plasmon resonance (SPR) and pull-down assays, indicating that PDLP7 might regulate PD function via a sphingolipids-mediated manner [25]. Firstly, we observed the precise localization of the PDLP7 protein on PDs via immunogold labeling transmission electron microscope (TEM). Validated PDLP7 polyclonal antibody [25] was used to indicate the distribution of PDLP7 in isolated PD fractions and leaf cell samples from four-week-old *Arabidopsis* wild-type (WT) plants individually. The result showed that gold particles (indicating PDLP7 protein) were deposited in the isolated PD fractions (as vesicle forms) (Figure 1A–C), scattered along the cell membrane (Figure 1D–F, red arrows; Appendix A), and aggregated around PDs (Figure 1D–F, red dashed boxes; Appendix A), but were not detected in the cytoplasm (Appendix A). Interestingly, we noticed that the distribution of these PD proteins changed with the expression level. Confocal observations revealed that PDLP7 driven by its native promoter exhibited a distribution consistent with that of most PD-localized proteins, primarily showing punctate fluorescence between the plasma membrane (PM) (Appendix A). In contrast, surface plots revealed that excessive expression of PDLP7 protein (35S:PDLP7) in the cells caused its extension from the intercellular junctions (Figure 1G, position 1; Figure 1H) to the cell membrane (Figure 1G, position 2; Figure 1I and Appendix A). By co-localizing PDLP7-RFP with callose, a PD marker, we also discovered significant overlap (Figure 1J), confirming that the punctate fluorescence of PDLP7-RFP is localized at the PDs.

Proteins containing GPI motifs are typically localized within detergent-resistant membrane (DRM) regions that are rich in sphingolipids and sterols [26]. The interaction between PDLP7 and the GPI-anchored BG10 indicated that PDLP7 may be anchored to the PD lipid rafts via sphingolipids, thereby regulating the plasmodesmata functions [25]. To verify our hypothesis, we extracted the PD DRM and obtained a gray-white flocculent substance containing the insoluble fraction after detergent (Triton X-100) treatment (Figure 1K). Western blot analysis displayed that PDLP7 were highly enriched at the DRM layer in the sucrose density gradient system (Figure 1K, right panel), indicating that PDLP7 was likely localized within the PD lipid raft regions. Previously, we reported that PDLP7 could bind with phytoshingosine (t18:0) in vitro [25]. Detailed molecular docking confirmed the interaction and showed the key residues of PDLP7 were LEU^271^, PHE^275^, and PHE^278^ in its single transmembrane region, with a binding free energy of −3.6 kJ/mol (Figure 1L). The three interacting residues overlapped considerably with the sphingolipid-binding motif (SBM) of PDLP7-STMD (the single transmembrane region shown in yellow; the structural simulation interaction sites highlighted in purple, Figure 1L). Furthermore, sphingolipidomic analysis of the extracted PD fraction revealed that the glycosyl inositol phosphorylceramides (GIPCs) was significantly reduced in the *pdlp7* T-DNA insertion mutant (from ~80% to ~60%, Figure 1M and Appendix A) compared with the wild-type (WT), hinting PDLP7 not only bind to sphingolipids, but might also affect the composition of sphingolipids. Overall, our results demonstrate that PDLP7, enriched in PD lipid raft nanodomains, may regulate PD function via lipids by modulating the dynamics of sphingolipids.

### 2.2. PDLP7 Could Form a Homologous Polymer in the Presence of Sphingosine

To test whether PDLP7 could form dimer, we used size exclusion chromatography (SEC) to detect the self-aggregation of the purified PDLP7 protein in vitro (Figure 2A and Appendix A). According to Cytiva Superdex 200 specifications (Appendix A), Aldolase (158 kDa) eluted at 12.5 mL post-sample loading (PSL, blue dashed line in Figure 2A), Ovalbumin (44 kDa) at 15 mL PSL (black dashed line), and Conalbumin (75 kDa) at 13.75 mL PSL (green dashed line). For purified His-TF-PDLP7 protein (theoretical MW: 81 kDa), the obvious single peak (closer to the baseline of a 75 kDa standard protein, Figure 2A) indicated that this protein mainly exists in a monomeric form. However, in vivo interaction assays showed that PDLP7 can self-interact (NuBG-PDLP7 + PDLP7-Cub Figure 2B, upper panels), as evidenced by growth on Synthetic defined (SD)–AHLT medium comparable to the positive control (NuBI + PDLP7-Cub Figure 2B, middle panels). Additionally, the combination of nLUC:PDLP7 and cLUC:PDLP7 resulted in luciferase expression at the designated injection site (Figure 2C, left panel), while the combination of nLUC:glu44 and cLUC:PDLP7 (negative control) produced no detectable signal (Figure 2C, right panel).

These results showed that PDLP7 existed as a monomer in vitro but formed homologous polymer in vivo, promoting us to speculate whether this protein might form oligomers in vivo with the assistance of lipids. Thus, we prepared proteoliposomes via embedding His-TF-PDLP7 into either 1-palmitoyl-2-oleoyl-sn-glycero-3-phosphocholine (POPC) (negative control) or POPC + t18:0 liposomes and utilized Bis(sulfosuccinimidyl) suberate (BS^3^) as a crosslinker to analyze the polymeric state of PDLP7 (Figure 2D). Western blot showed that the recombinant His-TF-PDLP7 showed no dimerization in the absence of liposomes or in the presence of only POPC liposomes (Figure 2D, left and right panels). However, in the POPC + t18:0 liposomes, the PDLP7 dimers (~160 kDa) was detected with prolonged cross-linking time (Figure 2D, middle panel). These results indicate that PDLP7 exists as a monomer in vitro, but the presence of sphingosine can induce its self-aggregation.

### 2.3. PDLP7 Regulates the Membrane Liquid-Ordered/Disordered Phases Through Interaction with Sphingolipids

Lipid rafts, enriched in sphingolipids and sterols, were relatively liquid-ordered membrane nanodomain [13]. We stained the WT and *pdlp7* root cells with di-4-ANEPPDHQ to detect dipole potential changes in the lipid bilayer. The dye is excited at 488 nm, with a peak emission wavelength of ~560 nm in the ordered phase and ~620 nm in the disordered phase (Figure 2E). Compared with the WT plants (−0.474 ± 0.038, Figure 2F, left panel and Figure 2G), di-4-ANEPPDHQ displayed a significantly lower generalized polarization (GP) value in *pdlp7* plants (−0.521 ± 0.031, Figure 2F, middle panel and Figure 2G) and significantly higher GP value in *PDLP7* overexpression line (−0.389 ± 0.053, Figure 2F, right panel and Figure 2G). These findings suggest that the membrane nanodomains in the *pdlp7* mutant were less packed and more disordered, whereas increased PDLP7 expression promoted lipid packing. This preliminary data suggests that the accompanying PDLP7 homopolymers may interact with PD/PM lipid rafts through the sphingolipid-binding motif to regulate the plasmodesmata function.

### 2.4. PDLP7 Interacted with PDLP5 In Vivo

PDLP5 also contained a sphingolipid-binding site in its TMD, and we further investigated whether PDLP5 and PDLP7 (both containing sphingolipid-binding motifs), would exhibit interactions (Figure 3). Pull-down assay showed there were no interaction between recombinant protein MBP-PDLP5 and His-TF-PDLP7 (Figure 3A). However, similar with PDLP7, the interaction between PDLP5 and PDLP7 could be detected in Bimolecular luciferase complementation (BiLC) and membrane yeast two-hybrid (MY2H) assays (Figure 3B,C). In *Nicotiana* cells, the combinations of nLUC:PDLP5 + cLUC:PDLP7 and nLUC:PDLP5 + cLUC:PDLP5 produce luciferase at designated areas (Figure 3B). In yeast cells, similar to the positive control NuBI + P5-Cub, the combinations NuBG-P5 + P5-Cub, NuBG-P5 + P7-Cub, and NuBG-P7 + P5-Cub all exhibit growth in Synthetic dextrose minimal medium (SD)–AHLT deficient medium (Figure 3C). These were verified by our previously published PDLP7 co-immunoprecipitation (CO-IP) proteomics datasets [25], where PDLP5 was detected in all three independent replicates. In one replicate, both the number of peptides and the number of unique peptides exceeded two (Appendix A). These experiments collectively suggest that although PDLP5 does not interact with PDLP7 in vitro, interactions are evident in vivo.

### 2.5. The Overexpression of PDLP5 and PDLP7 Leads to Upregulation of SA-Characterized Gene PR1 in the Cells

Typically, in normal circumstances, PDLP5 and PDLP7 are expressed in different cells or PDs. Notably, *PDLP7* exhibited substantially higher expression in the leaf petiole compared to *PDLP5*, as verified by qRT-PCR (Figure 4A). Additionally, the β-galactosidase (GUS) signal preferentially stained the veins and petioles in *proPDLP7:GUS* lines (Appendix A). Although PDLP5-mCherry and PDLP7-GFP exhibited fluorescence at the plasma membrane and PDs, they displayed only partial co-localization. For instance, in the right panel of Figure 4B, at arrow ‘a’, the two proteins demonstrate strong co-localization, whereas at ‘b’, PDLP5 and PDLP7 do not co-localize (Figure 4B,C and Appendix A). We further analyzed the fluorescence localization of interactions through Bimolecular fluorescence complementation (BiFC). When only PDLP5 is overexpressed (P5-nYFP + P5-cYFP), the yellow fluorescence exhibits strong co-localization with PD marker callose, yielding a co-localization coefficient of 0.79 ± 0.02 (Figure 4D, upper panels and Appendix A). In contrast, when both PDLP5 and PDLP7 are overexpressed (P5-nYFP + P7-cYFP), some punctate YFP fluorescence accumulated at the plasma membrane, and while it still colocalized with callose, the Pearson’s *R* value was significantly decreased compared to the overexpression of PDLP5 alone (Figure 4D, middle panels; Appendix A). Although the punctate yellow fluorescence at the plasma membrane still co-localized with callose, the presence of large intracellular fluorescent aggregates considerably reduced the Pearson’s *R* value (0.47 ± 0.07, Figure 4D, lower panels and Appendix A).

We observed that co-expression of PDLP5 together with PDLP7 promoted a leaf cell in atrophy state, in which the cell membrane partially departed from the cell wall in the *Nicotiana* (*Nt*) system (Figure 4D, lower panels). In addition, gene *pathogenesis-related protein 1* (*NbPR1*), which was one of the salicylic acid (SA) marker genes, was highly upregulated via qRT-PCR, compared with the mock plants and even overexpression of PDLP5 (Figure 4E). Additionally, in *Arabidopsis* plants, we also verified this conclusion via transiently expressing the P5-nYFP + P7-cYFP combination. The qRT-PCR results demonstrated that, relative to the mock plants, the *AtPR1* and *AtNPR1* were significantly upregulated in the PDLP5-PDLP7 combination, while most *PLANT DEFENSIN (PDF)* genes did not exhibit upregulation during this process (Appendix A). To determine the biological significance of the simultaneous overexpression of PDLP5 and PDLP7, it prompts us to consider that following *turnip mosaic virus* (TuMV) infection, the transcriptional levels of *PDLP5* and *PDLP7* are significantly upregulated compared to mock-inoculated plants [25]. Consequently, we further utilized the aforementioned materials to conduct PDLP7 immunoprecipitation (IP) assays and measured the amount of PDLP5 that was pulled down by the IP (Figure 4F). The Western blot results indicate that, under the same amount of PDLP7 (Figure 4F, lower panel), a greater quantity of PDLP5 is pulled down in the TuMV-infected material (grayscale: 0.52 vs. 0.25, Figure 4F, upper panel). This demonstrates that more PDLP5 and PDLP7 may interact with each other post-viral infection to confer resistance. These findings suggest that the simultaneous overexpression of *PDLP5* and *PDLP7* may lead to the upregulation of plant *PR1* genes, potentially triggering the activation of the SA pathway, but not the jasmonic acid (JA) pathway, thereby promoting antiviral immunity.

In summary, we propose a hypothetical model to explain how two structurally conserved yet functionally distinct plasmodesmata-located homologous protein coordinate PD permeability via regulating membrane lipid order in response to viral infection (Figure 5). Under normal physiological conditions, PDLP5 interacts with the GSL8 to control callose synthesis in source cells (like mesophyll tissues), whereas PDLP7 interacts with BG10 to modulate callose degradation in sink cells (like vascular tissues). Together, these two proteins establish a finely tuned regulatory network that dynamically controls PD callose turnover in different tissues or PDs (Figure 5A). When virus attacks, both PDLP5 and PDLP7 are upregulated and increasingly recruited to the PD membrane, where their interaction likely promotes a more packed lipid environment and induces salicylic acid-responsive genes like *PR1*, ultimately leading to PD tightly closure as part of the antiviral defense response (Figure 5B).

## 3. Discussion

### 3.1. Determinants of the Localization of PDLP Protein on PDs

Located at the PDs, PDLP family proteins contain one or two Domain of Unknown Function 26 (DUF 26 domains) in their extracellular region, a type I transmembrane region, and cytoplasmic tail [27]. Thomas also showed that the completeness of the TMD of PDLP proteins guaranteed its correct localization to PDs [27]. Evidence showed that the TMD sequence (amino acid composition or length) of each *Arabidopsis* PDLP member was essential for PDLPs to self-interact and anchor at the membrane, but not for the plasmodesmal targeting [28]. Recent studies have indicated that the juxta-membrane domain (JMe), an extracellular region with signaling function, also plays a crucial role in the localization of PDLPs to PDs [29]. Deletion of the C-tails did not affect the location, but whether it regulates its PD location efficiency is still unknown. Thus, it was hard to conclude that the structural features completely determined the PDLP specific location. In our case, we found that overexpressing PDLP7 or PDLP5 in *Nicotiana* leaf cells showed a distribution shift from PDs to PM (Figure 1G and Figure 4B). Specifically, PDLP1 has been reported to overflow from PDs to extra-haustorial membrane upon downy mildew pathogen *Hyaloperonospora arabidopsidis* infection [30]. However, the overexpression system has certain limitations, as PDLP7 and PDLP5 may mislocalize from PDs to PM. Future studies could incorporate native experimental data—for instance, examining whether PDLP7 upregulation following viral infection results in increased protein localization to PM. Integrating fluorescence localization techniques with cell biology experiments may offer additional insights. Thus, we proposed that the precise targeting of PDLPs to PDs might be due to comprehensive factors, including the structural characteristics, the expression level, and the microenvironment of PDs.

### 3.2. Correlation Between PDLP7, Lipidomics, and Lipid Order

As tiny structures extending from the PM, PDs are important for maintaining intercellular communication acting as unconventional membrane contact sites [31]. Lacking PDLP7 impairs the GIPC content in the PD fraction, which suggests that a stable PD membrane surrounding requires proteins or other factors to be maintained. The PDLP7 might act as a restraint to prevent GIPCs from flowing out. The GIPC content might participate in the formation of lipid raft domain, which might dramatically affect the membrane order, which might be indicated as GP values [32]. In our work, we found that PDLP7 affected the orderliness of the whole cell membrane (Figure 2E–G). In the *pdlp7* mutant, the PM of the root cells exhibited a more disordered state. GIPCs are the most abundant sphingolipid of PMs, which might enhance the sterol-induced ordering effect [32]. Thus, further studies are still required to examine the influence of PDLP7 on the total PM sphingolipid composition.

In addition, PDLPs also determine callose contents and affect the stress response of plant cells. For example, it has been reported that overexpression of PDLP5 may lead to an increase in the SA level in cells, and the increase in SA will greatly affect the order of the cell membrane [33]. Thus, the decreased membrane order might be caused not only by the reduced GIPC contents but also by callose or other factors. Therefore, understanding the correlation between PDLP7, lipidomics, and lipid order is essential for comprehending the complex interplay between cellular structures and functions.

### 3.3. Plant Cells Possess at Least Two Regulatory Models for Sphingolipid–PDLPs

Among the eight PDLPs, only PDLP5 and PDLP7 contain putative sphingolipid-binding sites and interacted with t18:0 [24,25]. PDLP5 directly interacted with GSL8, potentially regulating callose synthesis to control PD aperture [22], whereas our recent research indicated that PDLP7 regulates callose hydrolysis through its interaction with BG10, thereby modulating PD function [25]. This series of work indicated that plant cells might possess two sets of different PDLP–sphingolipid-mediated PD regulatory strategies. Single-cell transcriptomic data revealed that PDLP5 is primarily expressed in mesophyll cells, whereas PDLP6 and PDLP7 are expressed in vascular tissues, such as phloem parenchyma (PP) cells, leaf veins, and petioles [34]. Actually, vascular cells exhibit extreme specialization in types, particularly in PD structure and function. For example, PDs pass through mesophyll cells and phloem cells (M-to-P), and those linking the xylem cells and phloem cells (X-to-P) are dramatically different [35,36]. These specialized PDs can significantly facilitate the transport of sucrose from mesophyll cells to phloem cells. However, the lack of lipidomic analyses of M-to-P PDs and X-to-P PDs restricts our understanding of the formation processes of these distinct PD structures.

Using the 35S promoter in the *Nt* system for co-expression, we found that PDLP5 and PDLP7 showed very little colocalization in PDs (Figure 4D). This suggests that PDLP5 and PDLP7 were selected to function at their individual sites. We also observed aggregated fluorescent states in the PDLP5 and PDLP7 co-expressed leaf cells (Figure 4D). Thus, we suspect that the co-localization of the two PDLPs might be a plant stress signal. The high expression level of the SA-responsive gene *PR1* supported our hypothesis (Figure 4F and Appendix A). We also observed that PDLP5 and PDLP7 were significantly upregulated following TuMV infection, which may contribute to the closure of plasmodesmata to resist pathogen invasion. Additionally, during the TuMV attack, the interaction rate between PDLP5 and PDLP7 increased (Figure 4F). These results suggest that plant cells likely recruit PDLP5 and PDLP7 in excess to close PDs in response to stress.

In total, we elaborated in detail on the effects of sphingolipids on the PDLPs, specifically, their influence on the oligomeric state of these proteins on the membrane. Through localization experiments, we revealed that plant cells utilize at least two Sphingolipid–PDLP protein-related strategies to regulate the function of intercellular connections. This work further expounds on the precise regulation of plant cells in controlling PDs.

## 4. Materials and Methods

### 4.1. Plant Materials and Growth Conditions

*Arabidopsis thaliana* (Col-0) and *Nicotiana benthamiana* were cultivated in a fully automated walk-in greenhouse. The photoperiod was set to 12 h of light and 8 h of darkness, with a humidity level of 65% in a long-day environment. The *PDLP7* T-DNA insertion mutant (*SALK_035241C*) was obtained from the Nottingham Arabidopsis Stock Center (NASC) and has been previously verified for its homozygous. Unless otherwise specified, 3–4-week-old *Arabidopsis* or *Nicotiana* leaves were used.

### 4.2. Transient Transformation, Fluorescence Observation, and Callose Staining

The binary vectors (*35S:PDLP7:GFP*, *35S:PDLP5:mCherry*) were transformed into *Agrobacterium tumefaciens* GV3101 p19 competent cells (Weidi Biotech, Shanghai, China) via chemical transformation method. Specifically, approximately 100–200 ng plasmid DNA was introduced into competent cells, followed by a series of treatments: 5 min on ice, 5 min in liquid nitrogen, 5 min at 37 °C, and an additional 5 min on ice. Subsequently, the cells were allowed to recover at 28 °C for 2 to 3 h before being plated on a medium supplemented with the appropriate antibiotic. Correct clones were selected, grown overnight in 1 mL LB medium supplemented with appropriate antibiotics, and subsequently expanded in 5 mL LB medium. Cells were harvested by centrifugation (5000 rpm) and resuspended in infiltration buffer (10 mM MgCl_2_, 10 mM 2-Morpholinoethanesulphonic acid (MES), 100 μM Acetosyringone (AS)). Following a 3 h incubation, the suspension was injected into the abaxial side of 4-week-old *Nicotiana benthamiana* leaves. After 12 h of dark incubation, the plants were returned to normal conditions for 24 h, and fluorescence observation was conducted using a ZEISS 710 confocal microscope. Fluorescence was imaged using 488 and 543 nm excitation light for GFP and RFP excitation, respectively.

Callose was visualized using an aniline blue staining solution prepared by mixing 0.1% (*w*/*v*) aniline blue (Sigma-Aldrich, St. Louis, MO, USA) with 1 M glycine (pH 9.5) in a 2:3 (*v*/*v*) ratio and incubating the mixture at room temperature in the dark for 48 h. The samples were quickly immersed in the staining solution, which was supplemented with 1.5 mM 2-deoxy-D-glucose (DDG, Sigma-Aldrich) to inhibit de novo callose synthesis. The samples were vacuum-infiltrated in a desiccator and incubated in the dark for 10 min. The excitation wavelength for aniline blue is 405 nm, with an emission wavelength of 500–550 nm.

### 4.3. Isolation of PD Fraction and PD Lipid Raft

Approximately 30–40 g 3–4-week-old *Arabidopsis* leaves were thoroughly ground into a powder in liquid nitrogen and dissolved in cell wall preparation buffer (CWP, 100 mM Tris-HCl, 10 mM EDTA, 100 mM KCl, 0.45 M mannitol, 1 mM PMSF, 10% glycerol). The homogenate was incubated on ice for 20 min, followed by filtration through a 100 μm cell strainer to remove insoluble large particles. The homogenate was subjected to ultrasonic treatment twice using CWP and cell wall wash buffer (CWW, 10 mM Tris-HCl, 10 mM EDTA, 100 mM NaCl, 1 mM PMSF, 10% glycerol, respectively), with each treatment followed by centrifugation at 400× *g* for 10 min. The pellet gradually changed color from green to gray-white, indicating the enrichment of cell wall components.

Dissolve 1.4% (*w*/*v*) cellulase-R10 in the cell wall digestion buffer (10 mM MES, 4.4% mannitol, pH 5.5) to resuspend the cell wall components and incubate at 37 °C for 1–1.5 h to digest the cellulose. Following centrifugation (5850× *g*, 10 min) to remove undigested components, the supernatant was ultracentrifuged at 110,000× *g* for 1 h, and the pellet was resuspended in buffer to obtain the plasmodesmata (PD) fraction.

For lipid raft isolation, a total of 2 mg of the aforementioned PD protein was mixed with 300 μL of TED buffer (50 mM Tris-HCl, pH 7.4, 3 mM EDTA, 1 mM DTT) containing 10% Triton X-100 and incubated on ice for 30 min. A 65% sucrose-TED solution was pre-mixed with the above-treated PDs and layered into the bottom of an ultracentrifuge tube (approximately 4.29 mL). Subsequently, 1.43 mL of 48%, 35%, 30%, and 5% sucrose-TED solutions were carefully applied on top of this layer. The mixture was subjected to ultracentrifugation at 141,000× *g* for 20 h at 4 °C. The detergent-resistant membrane (DRM) fraction was collected between the 35–48% sucrose layers, specifically between the second and third sucrose layers. This fraction was extracted using a 1 mL syringe and tested by Western blots.

### 4.4. PD Sphingolipidomic Measurement

Approximately 100 μL of the PD fraction (containing ~200 μg protein equivalent) was added to 400 μL of deactivation buffer (0.01% [*w*/*v*] butylated hydroxytoluene in isopropanol) and incubated at 75 °C for 20 min. Then, 800 μL extraction solution I (chloroform:methanol = 1:1, *v*/*v*) was added to each sample with vortexing at room temperature. The mixtures were incubated for 24 h with shaking at 150 rpm at room temperature before being centrifuged at 3000× *g* for 20 min. Supernatants were dried in the SpeedVac rotary vacuum desiccator (Genevac, Ipswich, UK). Sphingolipids were dissolved in chloroform:methanol = 1:1 (*v*/*v*).

LC–MS/MS analysis of sphingolipids was conducted using an Exion UPLC-QTRAP 6500 Plus (Sciex, Framingham, MA, USA) LC–MS/MS system (curtain gas = 20, ion spray voltage = 5500 V, temperature = 400 °C, ion source gas 1 = 35, and ion source gas 2 = 35; Lipidall Technologies Company, Changzhou, China) equipped with a Phenomenex Luna 3 μm silica column (internal diameter 150 × 2.0 mm). Buffer A (chloroform:methanol:ammonium hydroxide = 89.5:10:0.5, *v*/*v*/*v*) and buffer B (chloroform:methanol:ammonium hydroxide:water = 55:39:0.5:5.5, *v*/*v*/*v*/*v*) were used as mobile phases. The program was set as follows: 0 to 5 min, hold at 95% buffer A; 5 to 7 min, decrease the gradient to 60% (*v*/*v*) buffer A; 7 to 11 min, hold at 60% (*v*/*v*) buffer A; 11 to 15 min, decrease the gradient to 30% (*v*/*v*) buffer A; 15 to 30 min, hold at 30% (*v*/*v*) buffer A; 30 to 35 min, increase the gradient to 95% (*v*/*v*) buffer A. Internal standards were purchased from Avanti Polar Lipids: D-erythro-sphingosine (Sph-d17:1) for sphingoids, N-heptadecanoyl-D-erythro-sphingosine (Cer-C17, d18:1/17:0) for Cers, N-(dodecanoyl)-1-β-glucosyl-sphing-4-ene (GluCer-C12, d18:1/12:0) for glycosyl ceramides, and ganglioside (GM1-d18:1/18:0-d3) for GIPCs.

### 4.5. Immuno-Electron Microscopy

Four-week-old *Arabidopsis* leaves were fixed in a 4% (*w*/*v*) paraformaldehyde solution prepared in 0.1 M phosphate buffer (pH 7.4). Fixation was carried out for 2 h at room temperature (RT), followed by an overnight incubation at 4 °C. After fixation, the samples were rinsed four times with 0.1 M phosphate buffer and then dehydrated using a graded ethanol series (30%, 50%, 70%, 85%, 95%, and 100%). The dehydrated specimens were infiltrated and embedded in LR Gold resin (Electron Microscopy Sciences, Hatfield, PA, USA #14370). The polymerization was conducted in a Leia AFS system under UV light, first at −20 °C for 24 h and then at room temperature for an additional 3 days. Ultrathin sections (75 nm thick) were prepared using a UC7 ultramicrotome and collected on single-slot nickel grids. Sections were incubated for 10 min in a blocking solution of PBS supplemented with 0.5% BSA-c (Aurion, #900.022) and 0.05% Tween-20. Subsequently, the sections were immunolabeled in a humidified chamber with an anti-PDLP7 antibody (catalog L0831A, produced by CUSABIO, Wuhan, China, https://www.cusabio.com/ accessed on 30 December 2025) diluted 1:5 (*v*/*v*) in blocking solution. Following five washes with PBS, the sections were incubated for 1 h at room temperature with goat anti-rabbit IgG (H&L) conjugated to 10 nm colloidal gold (Jackson ImmunoResearch, #111-205-144, diluted 1:30) as the secondary reagent. After immunolabeling, samples were processed for counterstaining with 2% uranyl acetate. Electron micrographs were then acquired at an accelerating voltage of 120 kV using a Tecnai G2 Spirit BioTWIN microscope (FEI, Eindhoven, The Netherlands) fitted with an Orius 832 CCD camera (Gatan, Inc., Pleasanton, CA, USA).

### 4.6. Di-4-ANEPPDHQ Staining

The Di-4-ANEPPDHQ dye operates on the principle of phase separation, effectively distinguishing between less packed and packed lipid environments, thereby facilitating the visualization of lipid nanodomains [13]. Seven-day-old *Arabidopsis* seedlings were stained with 5 μM Di-4-ANEPPDHQ dye (dissolved in DMSO) on ice for 5 min. After a simple wash with ultra-pure water, the samples were examined using ZEISS 710 confocal microscope with a 488 nm excitation wavelength and 500–580 nm and 620–750 nm receiving wavelength. The membrane lipid order was assessed using the generalized polarization value (GP value).

### 4.7. Molecular Docking of PDLP7 Protein and t18:0

The compound t18:0 was retrieved from PubChem (ID 122121). Its 2D planar structure was manually rendered using ChemDraw (veision 16.0.1.4) and converted into a 3D SDF structure file in Chem3D, then converted into PDB format in PyMOL (version 4.6.0). The three-dimensional structure of PDLP7 was retrieved from the AlphaFold database. Molecular docking analysis of PDLP7 and t18:0 was performed using the AutoDock tool (version 1.5.6), with all parameters set to default.

### 4.8. BiFC and BiLC

The full-length CDS sequences of target genes were constructed into the *pCambia1300-nLUC/cLUC* and *pCambia1300-nYFP/cYFP* vectors. The methods for *Agrobacterium* transformation, amplification, and injection were conducted as previously mentioned. For BiFC, fluorescence was observed using ZEISS 710 confocal microscope (Jena, Germany). The excitation wavelength for YFP was 488 nm, and the emission filter wavelengths were 505–530 nm for YFP. For BiLC, after spraying and applying the luciferase substrate (Vazyme, Nanjing, China) at the injection site, the luciferase signals were observed using a live imaging system (LB 985 NightSHADE, BERTHOLD technologies, Nanjing, China). The spectral peak was recorded at 560 nm, with exposure times ranging from 500 to 1000 s.

### 4.9. Dual-Membrane Yeast Two-Hybrid Systems

The CDS sequences of *PDLP5* and *PDLP7* were cloned into the bait vector *pPR3-N* and the prey vector *pBT3-SUC*, respectively. The NMY51 yeast competent cells (Weidi Biotech, Shanghai, China) were transformed with different combinations of plasmids. Add 2–4 μg of plasmid to the competent cells and incubate at 30 °C for 30 min, then transfer to a 42 °C water bath for 15 min. NuBI and NuBG were used as positive and negative controls, respectively. The transformed yeast cultures were then spread or titrated onto SD-Leu/Trp plates (as a growth control) and SD-Leu/Trp/Ade/His plates (for interaction detection).

### 4.10. Pull-Down Assay

*PDLP5* and *PDLP7* were constructed into the *pMAL-c5x* and *pCold-TF* vectors, respectively, resulting in the fusion of Maltose Binding Protein (MBP) and His-trigger factor (TF) tags. Approximately 20 mL of bacterial culture was induced, and the cells were collected by centrifugation at 5000 rpm for 10 min. The bacterial pellet was resuspended in pull-down protein lysis buffer (Thermo Fisher Scientific, Waltham, MA, USA) and subjected to sonication (2 s on/3 s off, 40% power, 10–20 min) to release the expressed proteins. Bacterial suspension was centrifuged at 13,000 rpm for 10 min, and the supernatant was served as the input for the pull-down assay. HisPur Ni-NTA beads (Thermo Fisher Scientific) were resuspended and washed with lysis buffer to remove ethanol. The His-TF-PDLP7 or His-TF bacterial suspension was incubated with the beads for 1–2 h to enrich the His-tagged proteins. Subsequently, the beads were washed 6 times with pull-down lysis buffer to eliminate non-specific adsorption. MBPs or PDLP5-MBPs were added in a combinatorial manner and incubated overnight with the beads. The protein levels in each component were detected using Western blot analysis.

### 4.11. Purification of His-TF-PDLP7

The constructs *pCold-TF-PDLP7* and *pCold-TF* empty vector were transformed into *E. coli* BL21(DE3) competent cells. The correct monoclonal strains were sequentially cultured in 1 mL and 5 mL of LB liquid medium, followed by large-scale expression in 1 L LB medium. When the OD_600_ reached 0.5–0.6, isopropyl-beta-D-thiogalactopyranoside (IPTG) was added to a final concentration of 0.8 mM, and the culture temperature was lowered from 37 °C to 16 °C for overnight induction. The cells were collected by centrifugation at 5000 rpm for 10 min, resuspended in His Binding Buffer (50 mM Tris-HCl, 300 mM NaCl, 20 mM imidazole, pH 7.5), and lysed by sonication (2 s on/3 s off, 40% power, ~1 h or until the lysate appeared clarified). After centrifugation at 13,000 rpm for 10 min, the supernatant was collected as the crude protein solution.

Ni-affinity purification of proteins was performed using the AKTA system. The crude protein extract was loaded on the Ni column at a flow rate of 3 mL/min. After equilibrating the column with His Binding Buffer, elution was performed using 20%, 60%, and 100% (*v*/*v*) His elution buffer (50 mM Tris-HCl, 300 mM NaCl, 300 mM imidazole, pH 8.0). The molecular weight and purity of the target protein were assessed using SDS-PAGE followed by Coomassie Blue staining (Mei5 Biotech, Beijing, China). Further purification of the Ni-purified protein was conducted by size exclusion chromatography on a Superdex 200 column (GE Healthcare, Uppsala, Sweden) equilibrated sequentially with ddH_2_O and phosphate-buffered saline (PBS) at 0.5 mL/min. The protein was concentrated to approximately 1 mL using an ultrafiltration centrifuge tube (Millipore, Cork, Ireland) and then loaded into a 1 mL loop sample ring with a syringe. The molecular weight and purity of the protein were determined based on the elution peak time (volume) and SDS-PAGE.

### 4.12. Preparation of Protein–Liposomes and Crosslinking

POPC and t18:0 (Avanti Research) were dissolved in chloroform (with dropwise methanol added to aid solubilization of t18:0), adjusted to a final lipid concentration to 1 mM, and mixed at a ratio of 0.95:0.05 (*v*/*v*) to prepare POPC + t18:0 liposomes. The Eppendorf tube was sealed with Parafilm, wrapped in aluminum foil, and vacuum-dried in a rotary evaporator for 60 min to remove chloroform completely. The dried film was rehydrated with 500 μL 10 mM HEPES-KOH buffer (pH 7.4), followed by the addition of 350 μL 10 mM HEPES-KOH buffer containing β-D-octyl glucoside. The suspension was gently mixed and incubated at room temperature in the dark for 10 min to ensure proper homogenization and dispersion of the lipid. Next, His-TF or His-TF-PDLP7 protein was adjusted to 2 μM, and 150 μL was added to the aforementioned mixture, followed by a 10 min incubation at room temperature in the dark. A total of 50 mg Bio-Beads SM-2 (Biorad, Hercules, CA, USA) were added to the proliposomes and incubated gently with rotation at 4 °C in the dark for 90 min, followed by an additional 350 mg of beads for another 90 min. After a 5 min resting step, the supernatant was collected to a new Eppendorf tube as the protein–liposome complex sample (final concentrations of approximately 200 nM protein and 1 mM lipid).

Crosslinking was performed using the chemical crosslinker bis-succinimidyl glutarate (BS^3^, Thermo Fisher Scientific, Waltham, MA, USA). BS^3^ was dissolved in HEPES/KOH buffer (10 mM, pH 6.2), and 1 μL of a 25 mM BS^3^ solution was added to 50 μL of the protein–liposome sample. The reaction mixture was initially incubated on ice for 5, 10, or 15 min, followed by a further incubation at room temperature for 5, 10, or 15 min. The reaction was terminated by the addition of 5 μL of 0.5 M Tris-HCl (pH 7.4) and continued to incubate for 15 min. Finally, 6 × SDS-PAGE sample buffer was added, and the mixture was heated at 95 °C for 5 min, then cooled to room temperature and stored at 4 °C for subsequent Western blot analysis.

## 5. Conclusions

We provided a comprehensive explanation of how PDLP7 interacts with sphingolipids to modulate plasmodesmata function. Moreover, via cytological methods and genetic techniques, we further revealed that PDLP7 and PDLP5 operate in a coordinated yet tissue-dependent manner to regulate intercellular communication during plant development and stress adaptation. These findings validate our initial hypothesis and support a precise regulation strategy: the plant cells utilized two different PDLP–sphingolipid-mediated PD regulation manners to fine-tune plasmodesmata dynamics and maintain the balance between growth and immune responses.

## Figures and Tables

**Figure 1 plants-15-00145-f001:**
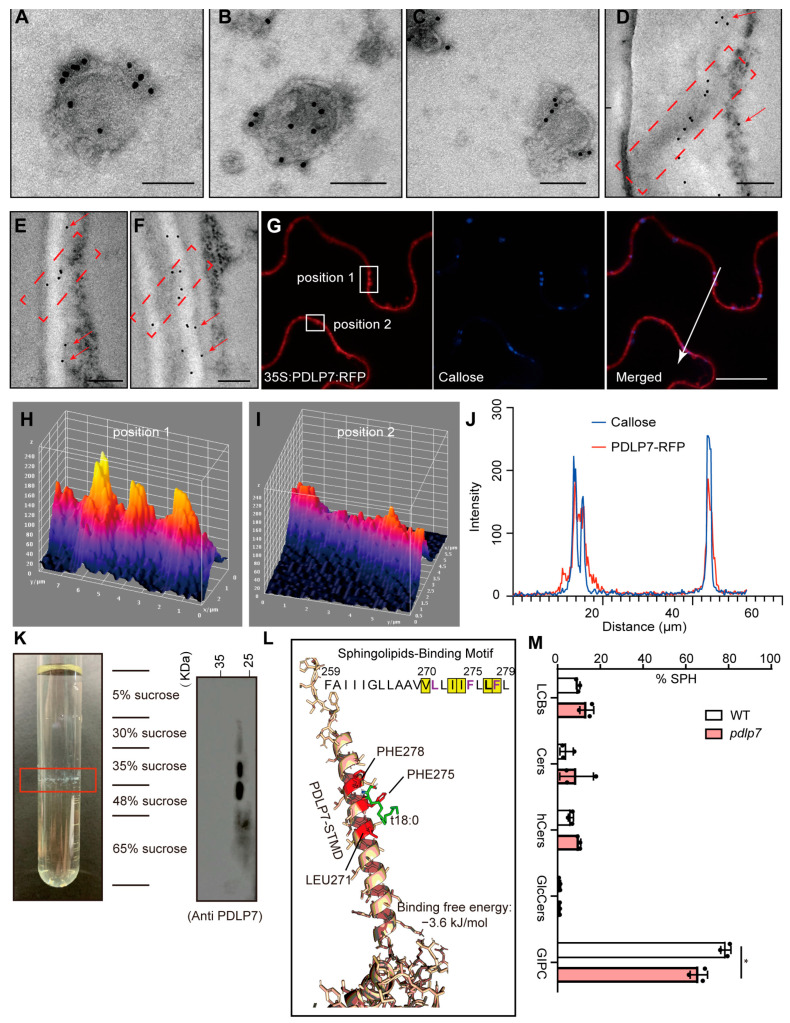
PDLP7 is located in the PD/PM and affects the composition of PD sphingolipids. (**A**–**C**) Immunoelectron microscopy was performed on the extracted plasmodesmata (PD) vesicle fraction using a PDLP7 polyclonal antibody. Gold particles were enriched in the PD vesicles. Scale bars: 100 nm. (**D**–**F**) PDLP7 was enriched in the plasma membrane (PM) and PDs. The red arrows indicate PM, and the red dashed box represented the PDs. Scale bars: 200 nm. (**G**) PDLP7-RFP co-localized with the PD marker callose in *Nicotiana benthamiana* epidermal cells at 2 d post-infiltration (dpi). Scale bar: 20 μm. (**H**,**I**) The protein distribution surface plot was generated from positions 1 and 2 (white solid boxes) in Figure 1G. (**J**) Co-localization curve analysis of PDLP7-RFP and callose. Fluorescence intensity along the region delineated by the white line was quantified using ImageJ (version 1.49v). (**K**) Separation of the PD lipid raft components using sucrose density gradient centrifugation. Photograph of the sucrose layers after centrifugation, with the red box indicating the 35–48% gray-white raft layer (**left panel**). Western blot analysis of gradient fractions using PDLP7 polyclonal antibody (**right panel**). (**L**) Molecular docking of the single transmembrane domain (STMD) of PDLP7 with the t18:0 molecule. (**M**) Sphingolipidomic analysis of PD components from wild-type (WT) and *pdlp7* mutant. Glycosylinositol phosphorylceramide (GIPC) content was significantly reduced in *pdlp7* mutant plants. Values reported represent the mean of 3 independent measurements, with error bars representing Standard Deviation (SD). Statistical significance was determined using Student’s *t* test. * *p* < 0.05.

**Figure 2 plants-15-00145-f002:**
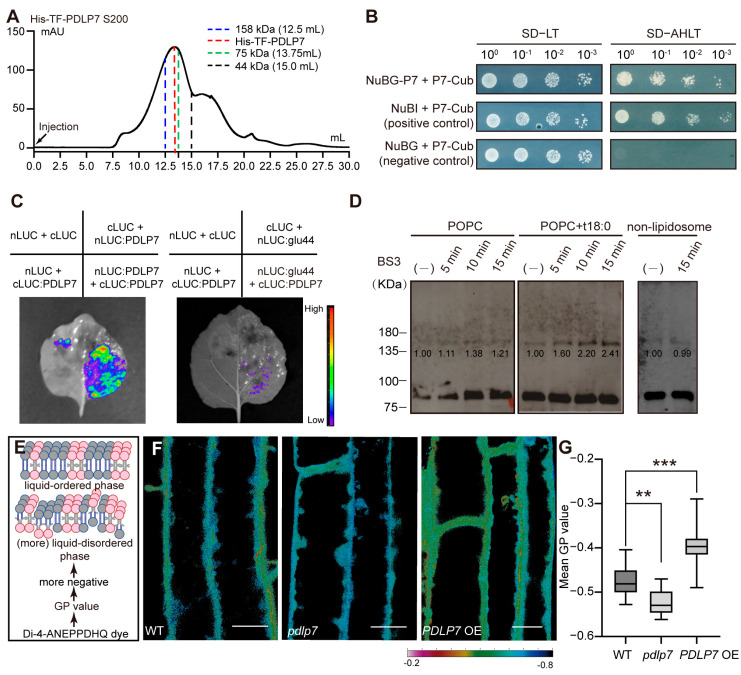
The PDLP7 protein exists as a monomer and forms a polymer with the assistance of sphingosine in liposomes in vitro. (**A**) The protein peak profile of His-TF-PDLP7 was analyzed using the Superdex 200. The protein was eluted between 12.5 and 15 mL after sample loading. (**B**) Dual-membrane yeast two-hybrid system corroborated that PDLP7 fused to NubG and Cub can undergo self-aggregation. The combinations of NuBI and NuBG with PDLP7-Cub served as positive and negative controls, respectively. (**C**) Co-expression of nLUC:PDLP7 and cLUC:PDLP7 produced luciferase signals at the designated injection sites, demonstrating that PDLP7 underwent self-aggregation in *Nicotiana benthamiana* cells. PDLP7 combined with Glu44 (At3g18080) served as a negative control. (**D**) Western blot analysis of PDLP7 in POPC and POPC + t18:0 liposomes. PDLP7 exhibited time-dependent dimerization in the presence of t18:0. (**E**) A schematic diagram illustrating the principle of observing lipid order in plasma membranes. (**F**) Fluorescence imaging of WT and *pdlp7* mutant cells stained with the Di-4-ANEPPDHQ dye. Scale bars: 10 μm. (**G**) Generalized polarization (GP) values were calculated to evaluate the lipid order in WT, *pdlp7* mutant, and *PDLP7* OE membranes. Data represent the mean ± SD of 12, 11, and 11 independent biological replicates, respectively. Statistical significance was determined using Student’s *t* test. ** *p* < 0.01, *** *p* < 0.001.

**Figure 3 plants-15-00145-f003:**
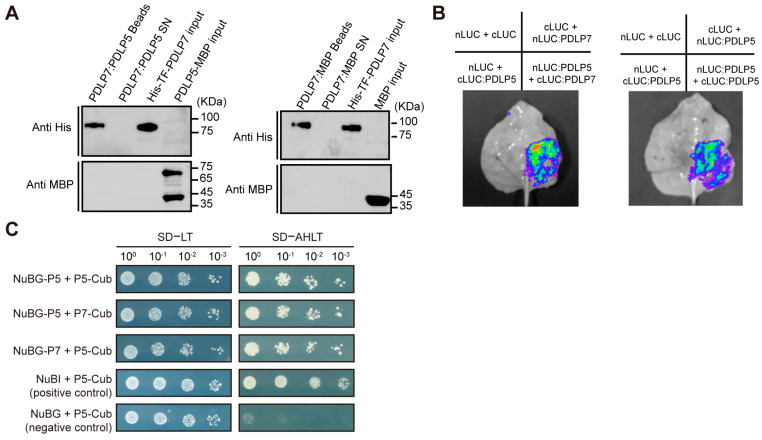
The PDLP7 protein interacts with PDLP5 within cells, forming heterologous dimers. (**A**) Pull-down assay demonstrated that PDLP5 and PDLP7 did not interact in vitro. PDLP5 and PDLP7 were fused with MBP and His-TF tags, respectively. His-TF or His-TF-PDLP7 was enriched using HisPur Ni-NTA beads, followed by incubation with MBP or MBP-PDLP5. Western blotting was used to detect the proteins in each component. (**B**) Bimolecular luciferase complementation assay confirmed the interaction between PDLP5 and PDLP7, as well as the self-aggregation ability of PDLP5. (**C**) Yeast two-hybrid experiments corroborated that PDLP5 could interact with PDLP7 and undergo self-aggregation. The combinations of NuBI and NuBG with PDLP5-Cub served as positive and negative controls, respectively.

**Figure 4 plants-15-00145-f004:**
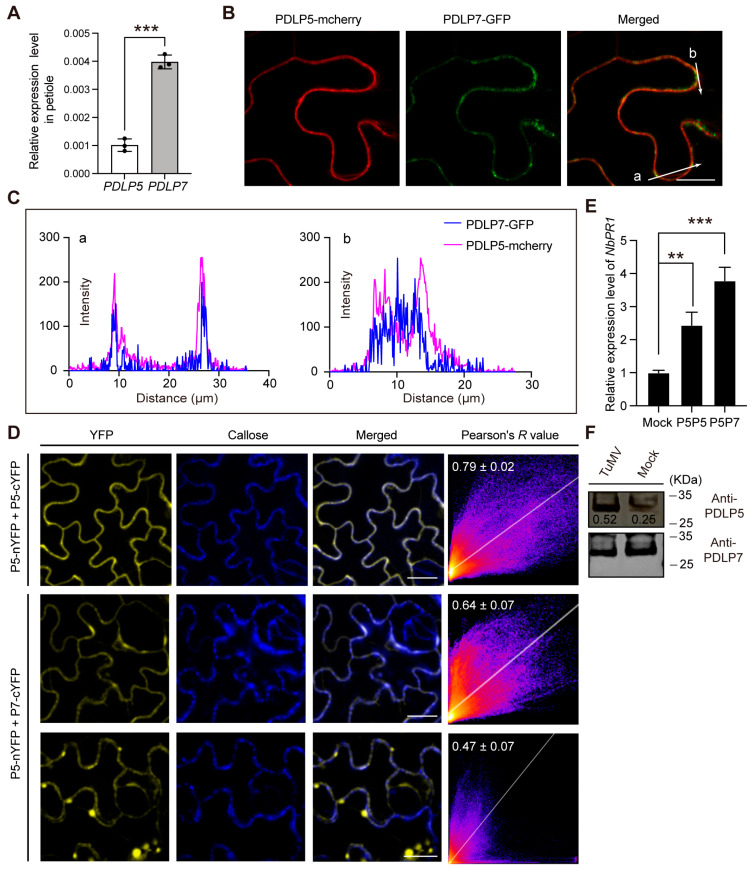
Simultaneous overexpression of PDLP5 and PDLP7 leads to partial loss of PD localization and upregulation of the SA-responsive gene. (**A**) Relative expression levels of *PDLP5* and *PDLP7* in leaf petiole determined by qRT-PCR. Data represent the mean ± SD of three independent biological replicates. (**B**) Co-localization analysis of PDLP5-mCherry and PDLP7-GFP in *Nicotiana benthamiana* epidermal cells at 2 dpi. Scale bar: 20 μm. (**C**) Fluorescence intensity analysis of the two marked positions (a and b) in (**B**), representing the co-localization and non-co-localization, respectively. The blue lines represent PDLP7-GFP, and the purple lines represent PDLP5-mcherry. (**D**) Bimolecular fluorescence complementation (BiFC) analysis of PDLP5 self-aggregation and the interaction between PDLP5 and PDLP7. Co-localization analysis was performed by combining P5-nYFP + P5-cYFP and P5-nYFP + P7-cYFP with callose, followed by analysis of the co-localization Pearson’s *R* value. PDLP5 showed clear co-localization with callose after self-aggregation, whereas co-overexpression of PDLP5 and PDLP7 partially lost the PD localization. Data represent the mean ± SD of three independent confocal microscopic views. Scale bars: 20 μm. (**E**) Relative expression of *pathogenesis-related protein 1* (*NbPR1*) by qRT-PCR after individual overexpression of PDLP5 or simultaneous overexpression of PDLP5 and PDLP7. Data represent the mean ± SD of three independent biological replicates. (**F**) Immunoblot analysis of proteins co-immunoprecipitated with PDLP7 (PDLP7-IP) from mock- or TuMV-infected leaves, probed with anti-PDLP5 and anti-PDLP7 antibodies. PDLP5 signal intensity was quantified relative to PDLP7 input levels and presented as a grayscale. Statistical significance was determined using Student’s *t* test. ** *p* < 0.01, *** *p* < 0.001.

**Figure 5 plants-15-00145-f005:**
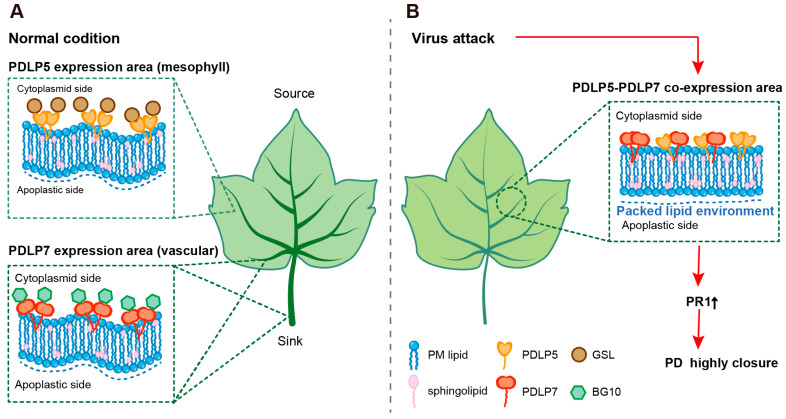
Proposed PDLP7- or PDLP5-mediated strategies to regulate PD function in plant cells. (**A**) Under normal physiological conditions, PDLP5 interacts with GSL8 to regulate callose synthesis in source cells (mesophyll), whereas PDLP7 interacts with BG10 to control callose degradation in sink cells (vascular). Collectively, these two proteins may orchestrate a sophisticated regulation of PD callose dynamics. (**B**) Upon viral infection, plant cells may rapidly recruit PDLP5 and PDLP7 to the PD membrane, where their interaction could induce a more packed lipid environment. This interaction is accompanied by the upregulation of the salicylic acid-responsive genes *PR1*, ultimately leading to PD closure as part of the defense response. The black arrow in the figure shows the upregulation of gene expression.

## Data Availability

The original contributions presented in this study are included in the article/Appendix A; further inquiries can be directed to the corresponding author.

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
