# Peer review of "Regulation of Plasmodesmata Function Through Lipid-Mediated PDLP7 or PDLP5 Strategies in Arabidopsis Leaf Cells"

_plants, 2026, doi:10.3390/plants15010145_

Round 1

Reviewer 1 Report

Comments and Suggestions for Authors

The article "Regulation of Plasmodesmata Function via Lipid-Mediated PDLP7 or PDLP5 Strategies in Arabidopsis Leaf Cells" by Xin Chen, Ning-Jing Liu, Jia-Rong Hu, Hao Shi, Jin Gao, and Yuxian Zhu presents an interesting, comprehensive study in which the authors model the formation of specific lipid modifications in Arabilopsis plasmodesmata.
The manuscript contains all necessary sections, is reliably and convincingly illustrated by various methods, and requires minor structural modification.
Minor comments: Please move the conclusions and results to the appropriate sections at the end of the introduction. Instead, formulate your intriguing hypotheses and tasks for testing their validity.
Be sure to reflect this in the conclusion, such as whether hypothesis 1 is confirmed, etc.
Also, please discuss the differences between different cell types and tentatively indicate the plasmodema varieties for which you believe this applies. I also don't see a number of references to modern reviews on this topic, which I think would be worth adding to the discussion specifically about the applicability of the lipid modification described by the authors during plasmodesmata formation. This may be especially important for cells of vascular tissues.
I recommend always using a bar to indicate the dimensions of cell images.
Also, indicate the type of tissue shown in the figure, as you used different models, including epidermis and vessels.
It is also common practice to show images in black and white to demonstrate that the cells are alive (dark field or other option). This is essential for vascular cells.
Indicate the leaf age, number of replicates, and describe the sample preparation of cytological preparations in more detail in the methodology.
After making changes, the aricle can be printed.

Author Response

Authors’ Responses to Reviewers’ Comments

Dear Editor and Reviewers,
We sincerely thank both reviewers for their constructive and helpful review comments and expertise the have put into the review, which has been invaluable in improving our manuscript. Below we provide point-by-point responses with revised text in red.

Reviewer 1
The article "Regulation of Plasmodesmata Function via Lipid-Mediated PDLP7 or PDLP5 Strategies in Arabidopsis Leaf Cells" by Xin Chen, Ning-Jing Liu, Jia-Rong Hu, Hao Shi, Jin Gao, and Yuxian Zhu presents an interesting, comprehensive study in which the authors model the formation of specific lipid modifications in Arabilopsis plasmodesmata.
The manuscript contains all necessary sections, is reliably and convincingly illustrated by various methods, and requires minor structural modification.

Minor comments: Please move the conclusions and results to the appropriate sections at the end of the introduction. Instead, formulate your intriguing hypotheses and tasks for testing their validity. Be sure to reflect this in the conclusion, such as whether hypothesis 1 is confirmed, etc.
[Author response] Thank you for your suggestions. We have removed the conclusions and result-oriented statements from the end of the Introduction. Instead, we now clearly present our central hypotheses and the specific questions to be addressed. These revisions are incorporated in the new paragraph at lines 86-93. In addition, we have updated the Conclusion section to explicitly reflect how our findings relate to the hypothesis proposed in the Introduction at lines 617-624.

Also, please discuss the differences between different cell types and tentatively indicate the plasmodema varieties for which you believe this applies. I also don't see a number of references to modern reviews on this topic, which I think would be worth adding to the discussion specifically about the applicability of the lipid modification described by the authors during plasmodesmata formation. This may be especially important for cells of vascular tissues.
[Author response] Thank you for your suggestion. We added a discussion about the possibility that different cell types might contain different types of PDs in lines 383-389. Additionally, we included more references.

I recommend always using a bar to indicate the dimensions of cell images.
[Author response] We have added the bars in Figures 2F, Figures 4B and Figures 4D. Thank you for pointing out the missing bars in those images.

Also, indicate the type of tissue shown in the figure, as you used different models, including epidermis and vessels.
[Author response] Thank you! We have indicated the tissue types in the figure legends and methodology.

It is also common practice to show images in black and white to demonstrate that the cells are alive (dark field or other option). This is essential for vascular cells.
[Author response] We have supplemented the bright-field images of Figure 1D, Figure 4B, and Figure 4D in the new Supplementary Figure 4 and 10. Thank you for your suggestion.

Indicate the leaf age, number of replicates, and describe the sample preparation of cytological preparations in more detail in the methodology.
After making changes, the aricle can be printed.
[Author response] Thank you for your suggestions.
(1) Leaves from 3-4-week-old Arabidopsis or Nicotiana were utilized for the isolation of the PD fraction, transient transformation, and immuno-electron microscopy assays etc. 7-day-old Arabidopsis seedlings were employed for Di-4-ANEPPDHQ staining. The leaf age information was detailed in the methodology section.
(2) We have added replicates such as lipidomics (lines 151-153), GP value (lines 211-212), qRT-PCR (line 288 and 299), and Pearson's R value (lines 296-297) in the figure legends.
Additionally, we refined the sample preparation process in the methodology section.

The manuscript presents valuable and original findings that merit publication after addressing the points raised above. The integration of lipidomics with plasmodesmatal biology is particularly novel and will be of significant interest to readers of Plants.

Reviewer 2 Report

Comments and Suggestions for Authors

The manuscript by Chen et al. presents a comprehensive and technically advanced study investigating how two plasmodesmata-localized proteins, PDLP5 and PDLP7, regulate plasmodesmatal function through sphingolipid-mediated mechanisms. The authors combine lipidomics, molecular biology, confocal microscopy, protein–lipid docking, protein biochemistry, and in vivo interaction assays to propose a mechanistic model in which PDLP5 and PDLP7 constitute two complementary regulatory modules that modulate callose turnover and lipid order at plasmodesmata, both during normal development and upon viral infection.

The topic is timely and of broad interest to the cell biology and plant physiology communities. The study provides new insights into the lipid–protein interactions that underlie plasmodesmatal specialization and significantly advances understanding of intercellular communication during stress responses.

The manuscript is generally well structured and clearly written. Experimental methods are detailed and reproducible. Figures are of high quality. The conclusions are supported by the presented data, however, several issues should be addressed to strengthen the manuscript before publication.

Suggestions:

The authors convincingly demonstrate that PDLP7 influences sphingolipid composition (notably GIPCs) and alters membrane order based on di-4-ANEPPDHQ staining. However, the mechanistic interpretation that PDLP7 regulates membrane order through direct modulation of lipid nanodomains remains somewhat speculative.

The authors should discuss alternative explanations, including indirect effects via altered callose dynamics or stress responses.

The manuscript would benefit from a more detailed correlation analysis between lipidomics, GP values, and PDLP7 expression levels.

High-level constitutive expression (35S promoter) leads to PDLP7 and PDLP5 mislocalization from PD to PM. This phenomenon is important but may reflect overexpression artifacts rather than biologically meaningful redistribution.

The authors should discuss limitations of the overexpression system and clarify whether native promoter lines display similar behavior under relevant physiological stimuli (e.g., viral infection).

I like also see more TEM (immunogold) photos - from Nicotiana.

The manuscript presents valuable and original findings that merit publication after addressing the points raised above. The integration of lipidomics with plasmodesmatal biology is particularly novel and will be of significant interest to readers of Plants.

Author Response

Authors’ Responses to Reviewers’ Comments

Dear Editor and Reviewers,
We sincerely thank both reviewers for their constructive and helpful review comments and expertise the have put into the review, which has been invaluable in improving our manuscript. Below we provide point-by-point responses with revised text in red.

Reviewer 2
The manuscript by Chen et al. presents a comprehensive and technically advanced study investigating how two plasmodesmata-localized proteins, PDLP5 and PDLP7, regulate plasmodesmatal function through sphingolipid-mediated mechanisms. The authors combine lipidomics, molecular biology, confocal microscopy, protein–lipid docking, protein biochemistry, and in vivo interaction assays to propose a mechanistic model in which PDLP5 and PDLP7 constitute two complementary regulatory modules that modulate callose turnover and lipid order at plasmodesmata, both during normal development and upon viral infection.
The topic is timely and of broad interest to the cell biology and plant physiology communities. The study provides new insights into the lipid–protein interactions that underlie plasmodesmatal specialization and significantly advances understanding of intercellular communication during stress responses.
The manuscript is generally well structured and clearly written. Experimental methods are detailed and reproducible. Figures are of high quality. The conclusions are supported by the presented data, however, several issues should be addressed to strengthen the manuscript before publication.

Suggestions:
The authors convincingly demonstrate that PDLP7 influences sphingolipid composition (notably GIPCs) and alters membrane order based on di-4-ANEPPDHQ staining. However, the mechanistic interpretation that PDLP7 regulates membrane order through direct modulation of lipid nanodomains remains somewhat speculative. The authors should discuss alternative explanations, including indirect effects via altered callose dynamics or stress responses.
[Author response] Thank you for your suggestion. We added this discussion in lines 366-372.

The manuscript would benefit from a more detailed correlation analysis between lipidomics, GP values, and PDLP7 expression levels.
[Author response] Thank you for your suggestion. We added more discussion between the lipidomics, GP values, and PDLP7 expression levels. Please see lines 354-372.

High-level constitutive expression (35S promoter) leads to PDLP7 and PDLP5 mislocalization from PD to PM. This phenomenon is important but may reflect overexpression artifacts rather than biologically meaningful redistribution. The authors should discuss limitations of the overexpression system and clarify whether native promoter lines display similar behavior under relevant physiological stimuli (e.g., viral infection).
[Author response] Thank you for your suggestions. We apologize for overlooking the potential artifacts that overexpression may introduce in our assessments. We attempted to extract plasma membrane (PM) fractions from both mock and TuMV-inoculated samples and employed Western blot analysis to determine whether PDLP7 relocates to the PM more frequently following viral inoculation. However, due to the substantial material requirements for plasma membrane extraction, we were unable to achieve reliable quantification. Consequently, we have acknowledged this limitation in the discussion and proposed potential future approaches, such as integrating cell biology and fluorescence observation experiments for further verification in lines 344-349.

I like also see more TEM (immunogold) photos - from Nicotiana.
[Author response] We have provided additional immunogold TEM images anti PDLP7, please see new Supplementary Figure 1. Thank you!

The manuscript presents valuable and original findings that merit publication after addressing the points raised above. The integration of lipidomics with plasmodesmatal biology is particularly novel and will be of significant interest to readers of Plants.

Round 2

Reviewer 2 Report

Comments and Suggestions for Authors

Thank you for correction of ms. I have only one suggestion, please give new Supplementary Figure 1 to main manuscript.

Author Response

Thank you for correction of ms. I have only one suggestion, please give new Supplementary Figure 1 to main manuscript.

[Author response] Thank you for your suggestion. We have provided new Supplementary Figure 1 to the main manuscript. Please refer to Figure 1A-F.